# HR-Extreme: A High-Resolution Dataset for Extreme Weather Forecasting

**Nian Ran**[1]**, Peng Xiao**[2]**, Yue Wang**[1]*__**Wenlei Shi**[3]**, Jiaxin Lin**[2]**, Qi Meng**[4]**, Richard Allmendinger**[5]*

[1]Zhongguancun Academy
[2]Hunan University
[3]Microsoft Research
[4]Chinese Academy of Sciences
[5]University of Manchester

## Abstract

The application of large deep learning models in weather forecasting has led to significant advancements in the field, including higher-resolution forecasting and extended prediction periods exemplified by models such as Pangu and Fuxi. Despite these successes, previous research has largely been characterized by the neglect of extreme weather events, and the availability of datasets specifically curated for such events remains limited. Given the critical importance of accurately forecasting extreme weather, this study introduces a comprehensive dataset that incorporates high-resolution extreme weather cases derived from the High-Resolution Rapid Refresh (HRRR) data, a 3-km real-time dataset provided by NOAA. We also evaluate the current state-of-the-art deep learning models and Numerical Weather Prediction (NWP) systems on HR-Extreme, and provide a improved baseline deep learning model called HR-Heim which has superior performance on both general loss and HR-Extreme compared to others. Our results reveal that the errors of extreme weather cases are significantly larger than overall forecast error, highlighting them as an crucial source of loss in weather prediction. These findings underscore the necessity for future research to focus on improving the accuracy of extreme weather forecasts to enhance their practical utility.

## 1 Introduction

Weather forecasting is a crucial scientific endeavor that influences various aspects of human life, from daily activities to disaster management and agricultural planning. The ability to predict weather conditions accurately can mitigate the impact of natural disasters, optimize resource management, and improve public safety. Traditionally, Numerical Weather Prediction (NWP) models (European Centre for Medium-Range Weather Forecasts (ECMWF), 2024; Bauer et al., 2015) have been the cornerstone of weather forecasting by explicitly solving the large-scale Partial Differential Equations (PDE). These models rely on mathematical formulations of atmospheric dynamics and physical processes to predict future states of the atmosphere based on current observations. While NWP models have achieved significant success, they are extremely computationally intensive, requiring substantial supercomputer power due to vast amount of data propressing and complex PDEs to simulate atmospheric conditions. Additionally, they have limitations in capturing the intricacies of certain weather phenomena.

In recent years, the advent of deep learning has revolutionized weather forecasting by offering alternative approaches that can potentially overcome some of the limitations of traditional NWP models by implicitly solving large-scale PDEs. Deep learning models such as Pangu-Weather (Bi et al., 2023), Fuxi (Chen et al., 2023b), and FourCastNet (Pathak et al., 2022) have shown promising results in generating high-resolution weather forecasts. FourCastNet is a Fourier-based neural network model, which leverages Adaptive Fourier Neural Operators and vision transformer (ViT) to produce high-resolution, global weather forecasts. Pangu-Weather leverages large-scale neural networks to

---

*Corresponding author,(yuewang@bjzgca.edu.cn, richard.allmendinger@manchester.ac.uk)

provide accurate and timely predictions. Fuxi integrates multi-scale spatiotemporal features to enhance forecast accuracy by effectively capturing the complex interactions within the atmosphere.

Despite these advances, one of the most critical and challenging aspects of weather forecasting is the accurate prediction of extreme weather events. Extreme weather, such as hurricanes, tornadoes, and severe storms, poses significant risks to life and property. Accurate forecasting of such events is essential for effective disaster preparedness and response. However, existing models and datasets often fall short in this regard. Current datasets lack sufficient types of extreme weather cases (Racah et al., 2017), or irrelevant data forms (Liu et al., 2023; Yu et al., 2023b; Zhu et al., 2016), or not very specific (Liu et al., 2021a), leading to models that are less reliable about extreme weather. On the other hand, current state-of-the-art (SOTA) deep learning models in weather forecasting have either been tested on only a limited range of extreme weather events or even lack testing in this area (Bi et al., 2023; Chen et al., 2023b; Pathak et al., 2022). This gap underscores the urgent need for specialized datasets and models that focus on extreme weather prediction.

In this work, we address this gap by proposing a comprehensive dataset that includes high-resolution extreme weather cases derived from the High-Resolution Rapid Refresh (HRRR) (National Oceanic and Atmospheric Administration (NOAA), 2024) data, a 3-km real-time dataset provided by the National Oceanic and Atmospheric Administration (NOAA). Our dataset, called HR-Extreme, includes high-resolution feature maps with dimensions (69, 320, 320) – where 69 represents the number of physical variables as channels and (320, 320) denotes the size of each feature map, with each pixel corresponding to a 3km by 3km area – aims to improve the accuracy of weather forecasting models in predicting extreme weather events. This dataset is built on U.S. area because of its complete and rich storm data. Our key contributions are:

- We propose a weather forecasting dataset called HR-Extreme specifically for evaluating extreme weather cases, based on HRRR data with a 3-km resolution, which provides a significantly higher resolution than previous most-used dataset ERA5, which is 31-km resolution, and is suitable for evaluating and improving SOTA deep learning and physics-based models.

- Our dataset includes a comprehensive set of 17 extreme weather types, such as strong winds, heavy rains, hail, tornadoes, and extreme temperatures, while previous work typically evaluates only one or a few types of extreme weather.

- We provide an extensive evaluation of SOTA models on our dataset, including visualization and detailed analysis. Additionally, we provided an improved version of deep learning model called HR-Heim based on HRRR data as a baseline, inspired by FuXi (Chen et al., 2023b) and MagViTv2 (Yu et al., 2023a), which outperforms SOTA methods on both normal weather forecasting and extreme weather evaluation in terms of one-hour prediction.

## 2 RELATED WORK

Historically, NWP models have been the gold standard, utilizing explicit solutions of PDEs to simulate atmospheric states. Although NWP models, such as those from the ECMWF, have achieved considerable success, they are computationally intensive and often struggle with the intricacies of complex weather phenomena.

### 2.1 ADVANCEMENT IN MODELS

Recently, weather forecasting has seen transformative advances with the integration of machine learning. Keisler (2022) utilizes graph neural networks to achieve successful short-term and medium-range forecasting on a $1.0°$ latitude/longitude grid. GraphCast (Lam et al., 2022) demonstrates that ML-based methods can compete with traditional weather forecasting techniques on ERA5 data, while ClimaX (Nguyen et al., 2023a) establishes a foundation model for weather prediction by pretraining and fine-tuning on several datasets using a transformer-based architecture (Vaswani et al., 2017). FengWu (Chen et al., 2023a) also employs a transformer architecture, addressing global medium-range forecasting as a multi-modal, multi-task problem and outperforming GraphCast. Subsequently, FengWu-GHR (Han et al., 2024) became the first data-driven method to successfully forecast at a $0.09°$ horizontal resolution. Stormer (Nguyen et al., 2023c) further

analyzes the key components that contribute to the success of transformers in this task. Additionally, GenCast and SEEDS (Price et al., 2023) emulate ensemble weather forecasts using diffusion models, enabling the generation of large ensembles that preserve the statistical properties and predictive skills of physics-based ensembles. In contrast, the NeuralGCM (Kochkov et al., 2024) model integrates a differentiable dynamical core for atmospheric dynamics with machine learning components, achieving significant computational efficiency and improved accuracy for both short-term and long-term predictions compared to traditional GCMs. Meanwhile, ClimODE (Verma et al., 2024) is a physics-informed neural ODE model that integrates value-conserving dynamics and uncertainty quantification, delivering state-of-the-art performance in both global and regional forecasting.

Models like Pangu-Weather (Bi et al., 2023) and FuXi (Chen et al., 2023b) represent significant strides in this domain. Pangu-Weather leverages three-dimensional neural networks with Earth-specific priors to effectively manage complex weather patterns, achieving higher accuracy and extended prediction periods. FuXi employs a cascaded machine learning system to provide 15-day global forecasts, demonstrating superior performance compared to traditional ECMWF models, particularly in extending the lead times for key weather variables.

Nevertheless, a critical challenge persisting in accurately weather forecasting is extreme weather events, which include hurricanes, tornadoes, severe storms and etc. These events pose significant risks to life and property, which makes precise forecasting essential for effective disaster preparation and response. However, recent advanced models, such as GraphCast, GenCast, FourCastNet (Pathak et al., 2022), Pangu, Fuxi, including physics-based models only involve a few types of extreme weather or only state the abilities in extreme weather prediction. Although NowcastNet (Zhang et al., 2023) is designed for extreme weather, it only predicts extreme precipitation. The reasons include insufficient attention to extreme weather events and the lack of a comprehensive and systemactic dataset that covers various types of extreme weather over a long period.

## 2.2 RELATED DATASET

Several datasets have been developed to address this challenge; however, they often fall short in comprehensiveness and resolution, or relevance. The ExtremeWeather dataset (Racah et al., 2017) is a notable effort, utilizing 16 variables, such as surface temperature and pressure, to detect and localize extreme weather events through object detection techniques and 3D convolutional neural networks (CNNs). However, this dataset is limited to three specific weather phenomena, such as tropical depression and tropical cyclones, and does not encompass a broad range of extreme weather types. Similarly, the EWELD dataset (Liu et al., 2023) focuses on electricity consumption and weather conditions, providing a temporal dataset that supports weather analysis over 15-minute intervals. However, EWELD provides time-series numerical data (signals) which is completely different from feature maps in HRRR data, and therefore it is not applicable in our weather forecasting problem.

ClimSim (Yu et al., 2023b) and ClimateNet (Prabhat et al., 2021) represent further advances in dataset development. ClimSim aims to bridge the gap between physics-based and machine learning models by providing multi-scale climate simulations with over 5.7 billion pairs of multivariate input and output vectors. Despite its scale, ClimSim is more geared towards hybrid ML-physics research rather than specific extreme weather forecasting and multi-channel feature map prediction. ClimateNet, on the other hand, provides an expert-labeled dataset designed for high-precision analyses of extreme weather events, such as tropical cyclones and atmospheric rivers. The most relevant work is a severe convective weather dataset proposed by Liu et al. (2021a), which includes ground observation data, soundings, and multi-channel satellite data for various types of extreme weather in China. Although extensive, their feature maps are of lower resolution, contain too much redundant data, involve fewer types of extreme weather, provide less accurate location information of extreme events, and different countries, compared to our dataset.

In conclusion, existing datasets often lack the necessary resolution and variety of extreme weather examples, which are essential for evaluating SOTA robust models. To address this gap, this study introduces a novel dataset derived from the High-Resolution Rapid Refresh (HRRR) data, a 3-km real-time dataset provided by NOAA. The HRRR dataset is significantly more detailed than previous datasets, offering high-resolution weather forecasting and a rich set of physical channels. This dataset is designed to enhance the predictive capability of weather forecasting with SOTA large deep learning models, particularly for accurately forecasting extreme weather events.

## 3 DATASET CONSTRUCTION

### 3.1 DATASET OUTLINE

In the context of weather forecasting, a model predicts a multi-channel feature map, where each channel represents a physical variable, such as temperature at two meters above ground, vertical wind speed, or humidity at a specific atmospheric pressure level. The dimensions of these feature maps depend on the resolution of the original data and the detection range. For example, each pixel in the feature maps in Pangu and Fuxi (Bi et al., 2023; Chen et al., 2023b) is 0.25° x 0.25°, constructing a grid of 721 × 1440 latitude-longitude points representing the global area. Although previous work has focused mainly on ECMWF ERA5 reanalysis data (Hersbach et al., 2020) (0.25° x 0.25° per pixel), our dataset is derived from HRRR (National Oceanic and Atmospheric Administration (NOAA), 2024) data provided by NOAA. This dataset features a 3-km resolution, is updated hourly, and offers cloud-resolving, convection-allowing atmospheric data with approximately ten times higher resolution than ERA5.

Our dataset comprises 17 types of extreme weather events that occurred in 2020, including extreme temperatures, hail, tornadoes, heavy rain, marine winds and etc. For each event, we specifically crop the area and utilize a series of feature maps with dimensions of (69, 320, 320) to cover both spatial and temporal axes. An event is thus expressed as $(n, t, c, w, h)$, where $n$ is the number of $320 \times 320$ feature maps covering the event area, $t$ is the timestamp, $c$ is the number of physical variables, and $w$ and $h$ denote width and height, respectively. In our case, we set $t = 3$ and $c = 69$, which means that for a particular hour when an extreme event occurs, we group the feature maps with this hour and two hours before. This approach accommodates models that take two previous timestamps of feature maps as input and predict the current timestamp. However, in our interface, users can adjust any number of hours before and after the timestamp for different usages.

The file naming convention used is "date_type1+type2+...typeN_minX_minY_maxX_maxY", for example "2020070100_Hail+Tornado_762_466_821_551.npz", where "date" is in the format "yyyymmddhh" and "type" denotes the event types included in this area, connected by "+". "minX, minY, maxX, maxY" are the indices of the area extracted from the original HRRR product, such that applying [minY:maxY, minX:maxX] on HRRR data will directly obtain this area. This format allows users to easily evaluate models in terms of different spans and event types, as well as visualizations. In the following sections, Section 3.2 will elaborate on how each type of extreme weather is collected and processed, and Section 3.3 will describe the creation of feature maps covering the area and the interface to generate additional years of data beyond current storage.

| Variable | Definition | Unit | Range |
|---|---|---|---|
| msl | Mean Sea Level Pressure | Pa | - |
| 2t | Temperature 2 m above ground | K | - |
| 10u | U-component Wind Speed 10 m above ground | m/s | - |
| 10v | V-component Wind Speed 10 m above ground | m/s | - |
| hgtn | Geopotential Height | gpm | |
| u | U-component Wind Speed | m/s | At 50, 100, 150, 200, 250, 300, |
| v | V-component Wind Speed | m/s | 400, 500, 600, 700, 850, 925, 1000 |
| t | Temperature | K | millibars, 13 levels in total |
| q | Specific Humidity | kg/kg | |

Table 1: Summary of the 69 physical variables in the dataset

### 3.2 DATA COLLECTION

Our dataset focuses on extreme weather events in the U.S. based on NOAA HRRR data. The original feature maps measure 1799 pixels in width and 1059 pixels in height, covering latitudes from 21.1° to 52.6° and longitudes from 225.9° to 299.1°, as collected by the Herbie Python library (Blaylock,

2024). The sources of extreme weather cases are divided into three categories based on extreme weather types.

**NOAA Storm Events Database** The first source is the NOAA Storm Events Database (National Centers for Environmental Information (NCEI), 2024), which includes a wide range of storms and significant weather phenomena from 1950 to 2024, recorded by NOAA's National Weather Service (NWS). While most events from this database include information on location and spatial range, some lack these critical details. Additionally, certain event types, such as avalanches and high surf, or events requiring long-term predictions like droughts, do not significantly impact the physical variables predicted by the models. To ensure greater accuracy, we have filtered out these events. Common types of extreme events not covered here will be supplemented by the other two sources. Otherwise the extensive and detailed records in this database facilitate the identification of areas and time spans through provided ranges and timestamps directly.

**NOAA Storm Prediction Center** The NOAA Storm Prediction Center (National Oceanic and Atmospheric Administration (NOAA), Storm Prediction Center (SPC), 2024) records daily reports of hail, tornadoes, and wind from various sources, including local NWS offices, public reports, agencies, and weather stations. However, the lack of event classification means that no specific ranges or time spans are provided, and multiple events are often mixed together. This complicates the creation of clear and accurate areas necessary for training and evaluating deep learning models accurately and effectively. To address this, we employ an unsupervised clustering algorithm to classify the events. After extensive case studies, we determined that DBSCAN (Ester et al., 1996) is the most suitable for this task, as illustrated in Figure 1. For each timestamp, user reports are treated as 2D points based on normalized latitude and longitude on the x and y axes. DBSCAN identifies clusters based on point density, forming a cluster if there are enough points in close proximity. We carefully tune the hyperparameters of DBSCAN to create more intuitive clusters and to filter out noisy points more accurately as shown in Figure 1. Noisy points are filtered out because they likely represent minor events or errors that are not significant enough to warrant creating a separate cropped area for evaluation.

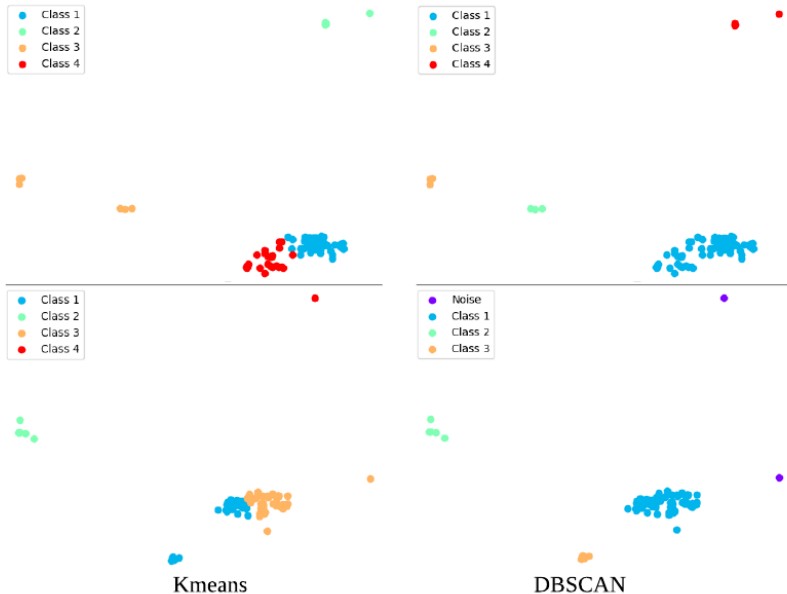

Figure 1: The performance of clustering by KMeans (right) and DBSCAN (left). It can be seen that the results of DBSCAN are more accurate and reliable, and the approach also identifies noisy points effectively.

**Manually Filtered Extreme Temperatures** As extreme temperature events are not covered in the Storm Events Database and are absent from the Storm Prediction Center, we manually created cases for unusual temperatures over large areas. ClimateLearn (Nguyen et al., 2023b) uses the 5th and

95th percentiles of the 7-day localized mean surface temperature for each pixel as a threshold to filter extreme temperature cases, similar to our approach. We also calculate the mean temperature for each pixel and use the 5th and 95th percentiles as a reference, and select the thresholds for extreme cold and heat at -29°C and 37°C, respectively, because exposure to such conditions can quickly cause discomfort or injury, and these temperatures are uncommon over large areas in the U.S. For each timestamp, we count the number of pixels in the feature map of the variable "temperature 2 meters above ground" that exceed these thresholds. If the count is below 100, we discard the timestamp since events in such timestamps are likely to be localized phenomena such as volcanic eruptions or fires rather than the widespread heat or cold events we aim to predict. Subsequently, we apply DBSCAN to each timestamp to further classify the events and eliminate noisy points.

### 3.3 DATA CROPPING AND INTERFACE

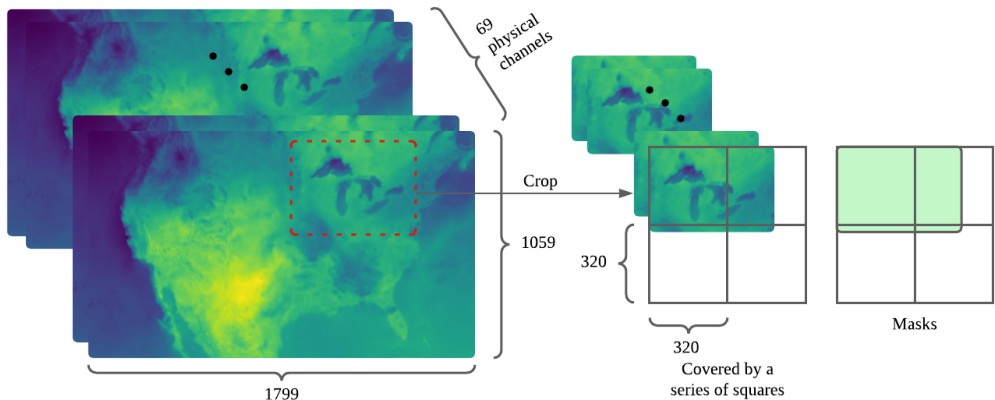

Figure 2: Event area are cropped and covered by a series of squares considering the fixed size fed to the neural network, masks are provided to ensure only event area are calculated.

Alongside the dataset generated by our code, we provide an index file that contains details on location, range, type of extreme weather, and time span by integrating information from above three data source. Users can easily convert this information from the index file to longitude and latitude coordinates with our open-source code. Based on the information from the index file, the original feature maps covering the US are cropped into a series of sub-images that specifically focus on extreme events. This approach allows models to train and evaluate on extreme events more precisely and exclusively. The entire pipeline is illustrated in Figure 2. To account for uncertainties in records and user reports, we slightly increased the range of each event. Each target area is covered by a series of squares with dimensions (320, 320) to ensure compatibility with neural networks, and masks are provided to ensure that only the target area is considered in loss calculations. When a cover image reaches the edge of the sampling image, it is moved completely inside the edges instead of adding constant value paddings, with masks adjusted accordingly. This ensures optimal performance of deep learning models, which are trained without constant padding for this specific task. We provide data for extreme events that occurred in 2020, totaling 5 TiB of memory. Data after July 2020 is designated as the evaluation set. Due to substantial memory requirements and varying user needs, we also offer a code interface for generating the dataset, allowing users to create datasets for any selected year, with any number of hours before and after for each piece of data, ensuring scalability and compatibility for different prediction strategies.

### 3.4 DATA AVAILABILITY

The HRRR data is in U.S. Government Work license, which means that the data is in the public domain and can be freely used, distributed, and modified without any restrictions. Our dataset (`https://huggingface.co/datasets/NianRan1/HR-Extreme`) and code (`https://github.com/HuskyNian/HR-Extreme`) will be available upon the accpetance of this paper.

# 4 EVALUATION

## 4.1 MODELS

**NWP** Numerical Weather Prediction models make predictions on HRRR data by solving mathematical equations that describe atmospheric dynamics and thermodynamics (Olson et al., 2022; Bauer et al., 2015). In this work, if refers to the WRF-ARWv3.9+ model (Skamarock et al., 2008) in HRRRv4. They assimilate observational data to create an initial atmospheric state, then use this data to initialize model variables. The model integrates these equations forward in time to produce forecasts of various atmospheric parameters at high resolution.

**Pangu** Pangu-Weather utilizes a 3D Earth-specific transformer (3DEST) architecture based on Swin-Transformer (Liu et al., 2021b). The 3DEST model incorporates height information as an additional dimension, enabling it to effectively capture atmospheric state relationships across different pressure levels. Predictions are made by processing reanalysis weather data through a hierarchical temporal aggregation strategy, which reduces cumulative forecast errors.

**FuXi** FuXi employs a cascaded deep learning architecture for weather forecasting. It features three components: cube embedding to reduce input dimensions, a U-Transformer using Swin Transformer V2 (Liu et al., 2022) blocks for data processing, and a fully connected layer for predictions. The cascade involves three models optimized for different time windows (0-5, 5-10, and 10-15 days), using outputs from shorter lead time models as inputs for longer ones to reduce forecast errors.

**HR-Heim** The architecture of HR-Heim follows a conventional structure with an encoder, a series of transformer layers, and a decoder, inspired by the FuXi architecture (Chen et al., 2023b). For the encoder, we utilize causal convolutions from MagViTv2 to capture spatial-temporal input (Yu et al., 2023a). The transformer segment consists of multiple stacked SwinTransformer blocks (Liu et al., 2022). Unlike typical Vision Transformer decoders that use a simple MLP with $1 \times 1$ convolution, which can hinder resolution, our decoder progressively upscales the feature map from $\frac{H}{h} \times \frac{W}{w}$ to the target size $H \times W$ through a series of steps. Each step resolves details at its specific resolution level, incorporating convolutional layers and upsampling operations to enhance prediction quality. More details about the architecture and its effectiveness are explained in the supplementary materials.

## 4.2 COMPUTE RESOURCES AND EXPERIMENT SETUP

The dataset creation process is resource-efficient, leveraging improved code efficiency and multi-threading capabilities. Generating the dataset for a half-year period requires approximately 8 hours on 42 CPU machines, utilizing around 2.5 TB of memory. HR-Extreme is designed to be machine learning (ML) ready, allowing users to simply load a file and use keys to retrieve inputs, targets, and masks to create tensors for a model. All models were evaluated on an Nvidia A100 80G GPU, with the evaluation of a deep learning model for half a year's data taking approximately 8 hours. For our experiments, we set the batch size to 8. Given HR-Extreme's ML-ready format, it is straightforward to use, with no additional parameter setup required.

## 4.3 EXPERIMENTS

| Model Type | RMSE on original set | RMSE on HR-Extreme | Mean increasement of RMSE of each variable |
|---|---|---|---|
| HR-Heim | **1.40** | **1.60** | **34.30%** |
| Pangu | 2.77 | 10.42 | 394.23% |
| Fuxi | 2.39 | 6.10 | 121.81% |
| NWP | 2.35 | 3.27 | 78.08% |

Table 2: Evaluation of NWP and deep learning models on extreme weather compared to on normal dataset.

All models (Pangu, Fuxi, and our HR-Heim) were trained on HRRR data spanning the U.S. from January 2019 to June 2020, from scratch. They were trained under identical parameters and same level of model parameters, and no hyperparameter tuning was applied to HR-Heim. Furthermore,

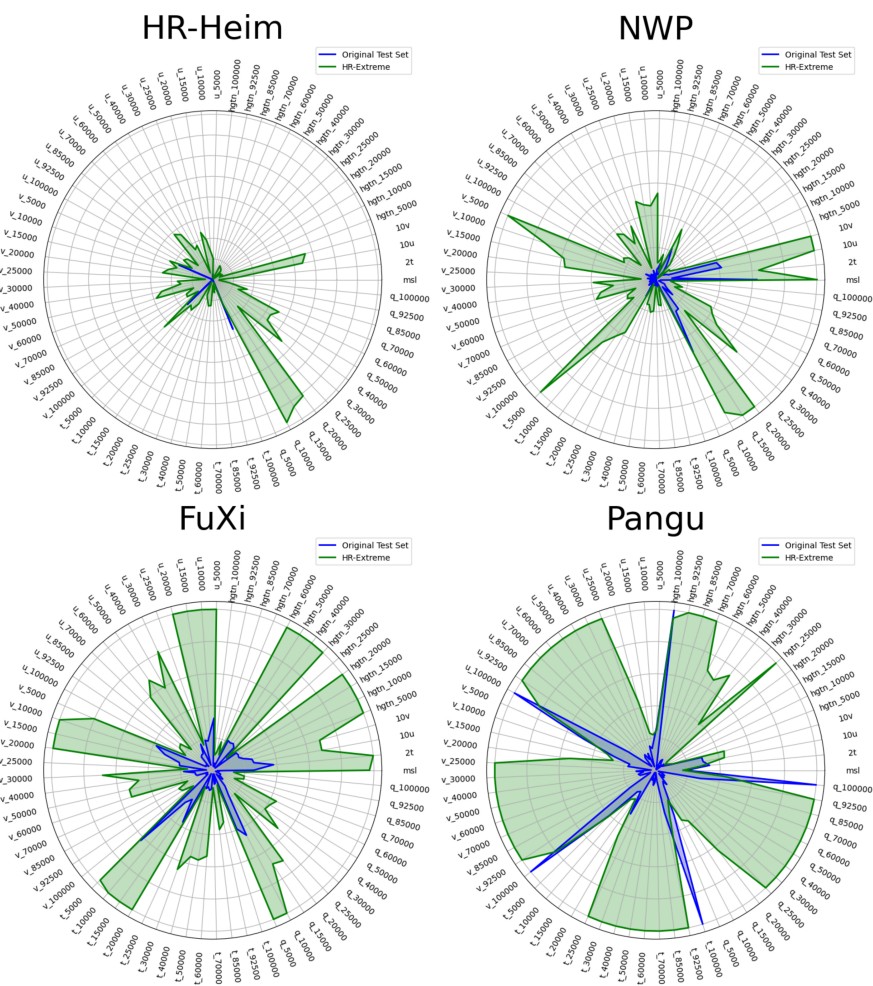

Figure 3: The comparison of each variable on original test set and HR-Extreme evaluated on four models, where the blue areas are normalized RMSE on original test set and green areas are normalized RMSE on HR-Extreme.

none of these models were fine-tuned on our HR-Extreme dataset, ensuring a fair basis for comparison and evaluation. We first evaluated the NWP model and the deep learning model on the original test set spanning from July 2020 to the end of 2020. Subsequently, these models were assessed and compared on HR-Extreme during the same period, as illustrated in Table 2. The losses on HR-Extreme for all models increased significantly compared to general losses. The highest increase was observed in Pangu, which experienced a nearly fourfold rise, while the smallest increase was noted in HR-Heim, at 34.3%. This underscores the substantial impact of extreme weather on model performance.

The complete loss for each variable, evaluated on both the original test set and HR-Extreme, and the comparisons are shown in Table 4 and Table 5 in the appendix A.2. The best results in these tables on the original test set are highlighted in black bold font, while those on HR-Extreme are highlighted in red bold font. In summary, HR-Heim exhibits superior performance on both the original dataset and HR-Extreme across nearly every variable. In terms of original test set, only 4 out of 69 variable losses are slightly higher than the results of NWP, and they are U and V component of wind speed, temperature and specific humidity at 50 millibars. On HR-Extreme, only the humidity at 50 and 100 millibars of HR-Heim are slightly higher than that of NWP and FuXi respectively, demonstrating HR-Heim as a strong baseline on HRRR data in terms of both general and extremes predictions.

From Figure 3, where the blue areas represent normalized RMSE on the original test set and the green areas represent normalized RMSE on HR-Extreme, it is evident that the losses for nearly all variables of all models on HR-Extreme increase significantly, often by many times compared to losses on the original set. However, HR-Heim performs most stably in both the original test set and HR-Extreme, with both green and blue areas of HR-Heim being dramatically smaller than those of other models. Notably, NWP shows more stability in terms of extreme weather detection compared to previous SOTA deep learning models, Pangu and FuXi, showing the demand of improving extreme weather predictions of deep learning models. We conducted additional experiments with

| Lead time | RMSE of NWP on original set | RMSE of NWP on HR-Extreme | RMSE of HR-Heim on HR-Extreme |
|---|---|---|---|
| 1 | 2.35 | 3.27 | **1.60** |
| 2 | 3.55 | 4.12 | **2.29** |
| 3 | 4.63 | 5.20 | **2.86** |
| 4 | 5.49 | 6.04 | **3.43** |

Table 3: RMSE of NWP and HR-Heim on HR-Extreme with different lead times

varying lead times for the NWP and HR-Heim models. The results show that losses gradually increase with longer lead times. Notably, the HR-Heim model consistently and significantly outperforms the NWP model across all lead times, achieving even lower losses than the NWP model on the original dataset. This demonstrates the HR-Heim model's effectiveness as a robust baseline for high-resolution predictions (e.g., HRRR) and extreme event analysis on HR-Extreme.

## 4.4 CASE STUDY

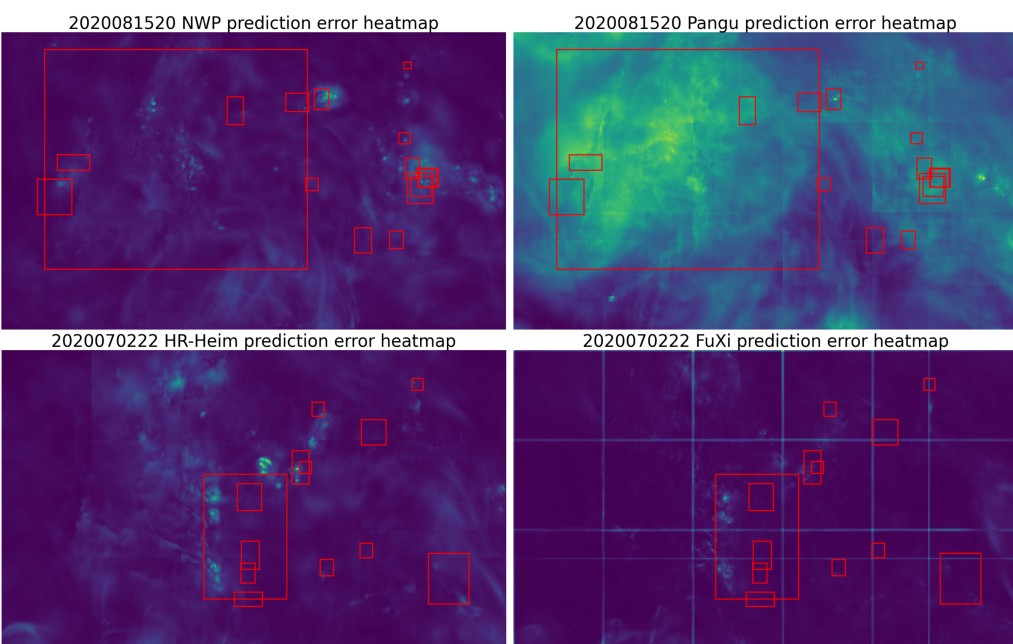

Figure 4: The mean error heatmap of all variables on entire U.S for four mdoels.

Figure 4 presents the mean error heatmap visualization of all variables with normalized loss across the entire U.S. Normalizing losses ensures each variable equally contributes to the error heatmap despite differing scales. The upper left and right panels depict NWP and Pangu predictions, respectively, at 8 p.m. on August 15, 2020, while the lower left and right panels show HR-Extreme and Fuxi predictions at 10 p.m. on July 2, 2020. Different models are used for a single timestamp

for comparison, and two timestamps are included for a more comprehensive demonstration. A red bounding box indicates the occurrence of a specific type of extreme event, and overlapping bounding boxes denote multiple extreme events in the same area.

In the upper left heatmap, bright areas covered with several bounding boxes indicate regions with high losses where multiple extreme weather events are occurring. The upper right heatmap features a very large bounding box with significant losses, which means Pangu makes significant errors in predicting this extreme event. Comparing the upper images highlights the superior generalizability of HR-Extreme, demonstrating its capacity to reveal the varying abilities of different models to detect extreme weather. Specifically, it shows that NWP's predictions for extreme events outperform Pangu's in this case, particularly regarding the largest bounding box, which corresponds to excessive heat. This comparison confirms that the most significant loss contributions are effectively identified by extreme weather.

The lower heatmaps further prove that the most prominent loss contributions are accurately situated within the bounding boxes. However, the most apparent loss areas are not always precisely centered within the bounding boxes, likely due to the movement of extreme weather events and HR-Extreme's basis on reports. Despite this, HR-Extreme exhibits excellent generalizability for both physical-based and deep-learning-based methods. We also provide more case studies of physical varibles in different extreme events in appendix A.1.

## 4.5 LIMITATION

Aside from excessive heat and cold extremes, HR-Extreme identifies bounding boxes primarily based on user reports, including inputs from individuals, weather stations, and agencies. This reliance introduces uncertainty in identifying the range and span of extreme weather events, particularly in regions with sparse populations or insufficient detection devices, such as mountains, marine areas, and deserts. Consequently, many events may be omitted. Although HR-Extreme encompasses a wide range of extreme weather events, it does not account for some large weather phenomena, such as tropical depressions and tropical cyclones(Racah et al., 2017). These phenomena are also crucial for predicting extreme weather events.

## 5 CONCLUSION AND FUTURE WORK

In this work, we present a comprehensive fine-grained dataset that encompasses 17 types of extreme weather events in the US, derived from HRRR data, which is updated hourly and features a 3-km resolution. Due to the incomplete records of extreme events in the NOAA database (National Centers for Environmental Information (NCEI), 2024), we utilized records from the NOAA Storm Prediction Center (National Oceanic and Atmospheric Administration (NOAA), Storm Prediction Center (SPC), 2024) and employed unsupervised clustering along with manual filtering to compile a complete set of extreme events that significantly impact HRRR feature maps. These events are anticipated to be accurately predicted by both deep learning and physical models, ensuring precise identification of extreme weather events. HR-Extreme serves as a novel dataset for evaluating the performance of weather forecasting models, particularly in practical applications. Our experiments indicate that the misprediction of extreme or "unusual" events significantly contributes to the overall prediction losses, revealing deficiencies in current model performance and underscoring the undervaluation of these events. Our proposed model outperforms SOTA transformers and NWP models in single-step prediction accuracy on both original dataset and extreme weather dataset.

For future work, since the current dataset is primarily built on user reports, developing more and improved methods combining with user reports to identify the time span and range of extreme weather events could yield a more accurate dataset. In addition, incorporating precursors of extreme weather events as well as large weather phenomenons such as tropical cyclones into the dataset could provide an even more comprehensive training framework. Furthermore, it would be beneficial to fine-tune models specifically on the extreme weather dataset to enhance their practical utility. Regarding HR-Heim, although it has superior performance compared to both SOTA deep learning models and NWP model, it is only trained and tested on single hour prediction, but it still serves as a strong baseline on HRRR data. Thus, an improved version of HR-Heim to predict longer period can be developed.

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

# A APPENDIX

## A.1 ADDITIONAL VARIABLE CASE STUDY

We conducted a statistical analysis comparing values between normal and extreme cases as shown in figure 5. To mitigate seasonal and regional effects, we randomly selected 50 timestamps within 15 days of the extreme event's start or end date, ensuring at least 24 hours separation from the event occurrence, within the same area. Extreme variables were averaged over the event duration, and details of the variables are provided in Table 1. For clarity, the analysis focused on variables closest to the ground. Extreme means differ significantly from normal cases, demonstrating the dataset's ability to statistically capture extreme values and events. Different types of extreme events exhibit distinct characteristics, while similar types show consistent features. During heavy rainfall, humidity increases rapidly, often with fluctuations in pressure and wind speeds. Thunderstorm winds display more complex variability in temperature, wind speed, and pressure. Cases of hail, lightning, and marine strong winds reveal different patterns: hail and lightning, which co-occur in this case, show significant increases in humidity, mean sea level pressure, and geopotential height, while marine strong winds show drops in mean sea level pressure and geopotential height, with a notable increase in humidity.

## A.2 DETAILED VARIABLE LOSS AND TYPE DESCRIPTION

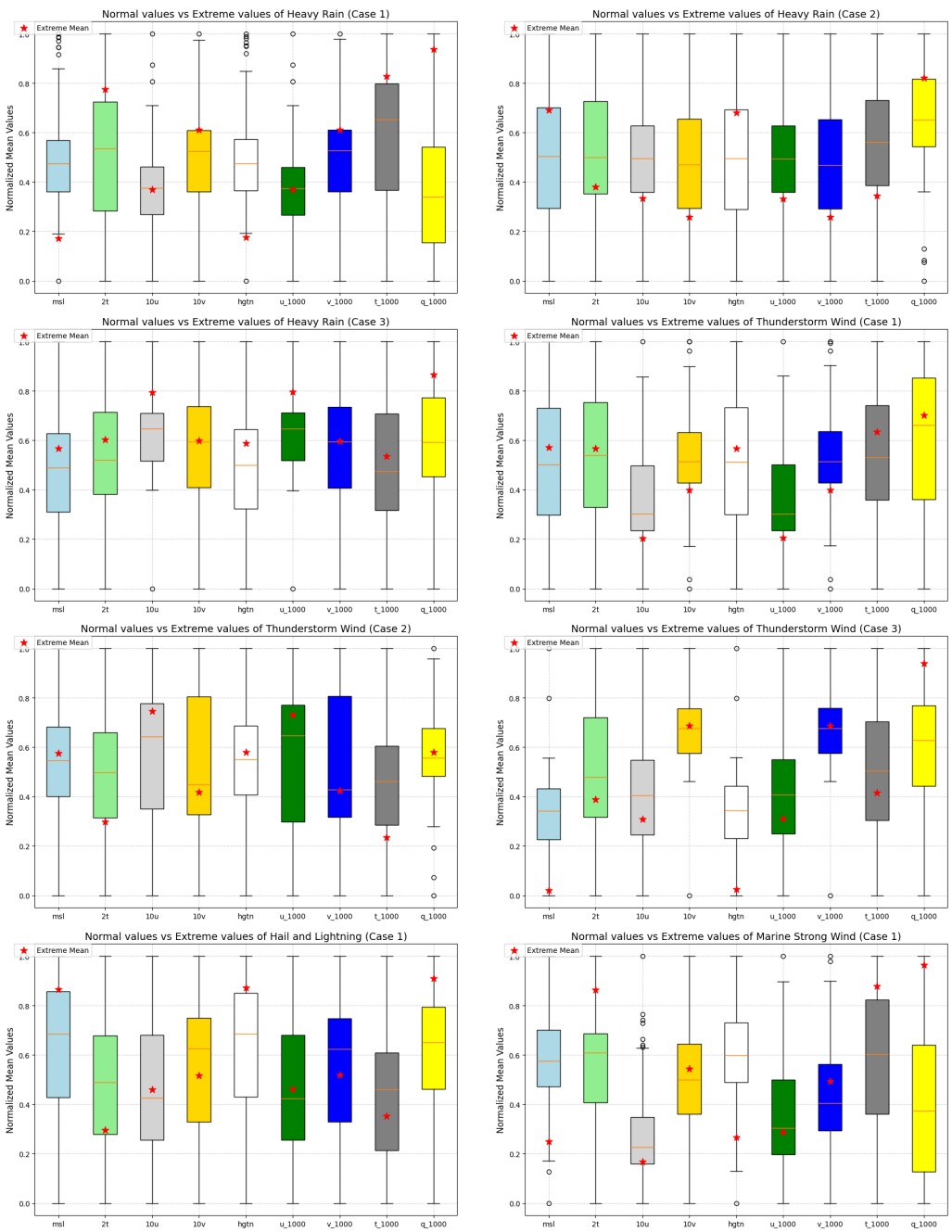

Figure 5: Case studies for physical variables in normal and extreme cases in the same area.

| Variable | HR-Heim General Loss | HR-Heim HR-Extreme Loss | Pangu General Loss | Pangu HR-Extreme Loss | FuXi General Loss | FuXi HR-Extreme Loss | NWP General Loss | NWP HR-Extreme Loss |
|---|---|---|---|---|---|---|---|---|
| msl | **27.00** | **27.88** | 37.09 | 36.06 | 40.48 | 78.61 | 60.30 | 80.07 |
| 2t | **0.40** | **0.47** | 0.81 | 0.74 | 0.86 | 1.63 | 0.52 | 1.19 |
| 10u | **0.59** | **0.75** | 0.81 | 0.90 | 0.78 | 1.09 | 0.89 | 1.32 |
| 10v | **0.59** | **0.77** | 0.82 | 0.93 | 0.78 | 1.12 | 0.89 | 1.36 |
| hgtn_5000 | **6.48** | **6.80** | 8.35 | 8.53 | 10.25 | 25.09 | 8.05 | 10.58 |
| hgtn_10000 | **5.32** | **5.93** | 6.87 | 7.39 | 9.54 | 28.76 | 7.04 | 8.56 |
| hgtn_15000 | **5.14** | **5.80** | 6.55 | 7.15 | 10.70 | 35.26 | 6.83 | 8.22 |
| hgtn_20000 | **4.77** | **5.54** | 6.77 | 7.53 | 10.54 | 36.72 | 6.50 | 8.12 |
| hgtn_25000 | **4.32** | **5.10** | 5.76 | 272.94 | 9.84 | 34.94 | 6.12 | 7.80 |
| hgtn_30000 | **3.95** | **4.69** | 5.19 | 14.26 | 9.01 | 30.41 | 5.82 | 7.38 |
| hgtn_40000 | **3.30** | **3.93** | 4.31 | 17.91 | 7.38 | 22.87 | 5.37 | 6.84 |
| hgtn_50000 | **2.85** | **3.39** | 4.06 | 13.11 | 5.95 | 18.27 | 5.18 | 6.73 |
| hgtn_60000 | **2.58** | **3.02** | 3.35 | 10.66 | 4.98 | 14.14 | 5.11 | 6.64 |
| hgtn_70000 | **2.40** | **2.69** | 3.05 | 83.58 | 4.21 | 10.82 | 5.06 | 6.42 |
| hgtn_85000 | **2.27** | **2.44** | 2.88 | 28.27 | 3.45 | 7.05 | 5.06 | 6.51 |
| hgtn_92500 | **2.31** | **2.44** | 3.24 | 39.93 | 3.40 | 6.46 | 5.15 | 6.86 |
| hgtn_100000 | **2.26** | **2.37** | 46.46 | 44.29 | 3.41 | 6.46 | 5.10 | 6.90 |
| u_5000 | 0.49 | **0.52** | 0.71 | 0.69 | 0.75 | 1.32 | **0.48** | 0.93 |
| u_10000 | **0.47** | **0.55** | 0.69 | 0.74 | 0.76 | 1.69 | 0.49 | 1.03 |
| u_15000 | **0.67** | **0.83** | 0.90 | 1.06 | 0.97 | 2.34 | 0.73 | 1.45 |
| u_20000 | **0.96** | **1.20** | 1.27 | 1.84 | 1.35 | 2.95 | 1.08 | 1.96 |
| u_25000 | **0.92** | **1.14** | 1.22 | 13.53 | 1.35 | 3.02 | 1.03 | 1.89 |
| u_30000 | **0.83** | **1.04** | 1.09 | 3.14 | 1.24 | 2.71 | 0.91 | 1.67 |
| u_40000 | **0.69** | **0.93** | 0.91 | 3.55 | 1.03 | 2.38 | 0.75 | 1.42 |
| u_50000 | **0.62** | **0.93** | 0.89 | 2.94 | 0.89 | 2.20 | 0.70 | 1.42 |
| u_60000 | **0.58** | **0.89** | 0.80 | 2.75 | 0.82 | 1.87 | 0.68 | 1.39 |
| u_70000 | **0.59** | **0.88** | 0.79 | 6.60 | 0.80 | 1.70 | 0.69 | 1.36 |
| u_85000 | **0.59** | **0.89** | 0.78 | 4.37 | 0.79 | 1.57 | 0.73 | 1.40 |
| u_92500 | **0.55** | **0.86** | 1.28 | 4.15 | 0.73 | 1.44 | 0.72 | 1.42 |
| u_100000 | **0.45** | **0.60** | 3.12 | 2.64 | 0.57 | 0.96 | 0.63 | 1.04 |
| v_5000 | 0.52 | **0.53** | 0.71 | 0.68 | 0.65 | 0.86 | **0.47** | 0.95 |
| v_10000 | **0.47** | **0.54** | 0.67 | 0.70 | 0.70 | 1.21 | 0.47 | 1.00 |
| v_15000 | **0.68** | **0.82** | 0.89 | 1.03 | 0.96 | 1.97 | 0.72 | 1.43 |
| v_20000 | **0.98** | **1.19** | 1.28 | 1.90 | 1.34 | 2.69 | 1.07 | 1.95 |
| v_25000 | **0.94** | **1.13** | 1.25 | 11.74 | 1.36 | 2.63 | 1.04 | 1.92 |
| v_30000 | **0.84** | **1.03** | 1.11 | 3.10 | 1.25 | 2.38 | 0.92 | 1.72 |
| v_40000 | **0.70** | **0.93** | 0.93 | 3.64 | 1.02 | 2.06 | 0.76 | 1.48 |

Table 4: Losses of each variable for each model

| Variable | HR-Heim General Loss | HR-Heim HR-Extreme Loss | Pangu General Loss | Pangu HR-Extreme Loss | FuXi General Loss | FuXi HR-Extreme Loss | NWP General Loss | NWP HR-Extreme Loss |
|---|---|---|---|---|---|---|---|---|
| v_50000 | **0.63** | **0.94** | 0.90 | 3.06 | 0.90 | 1.92 | 0.71 | 1.48 |
| v_60000 | **0.59** | **0.91** | 0.81 | 2.82 | 0.82 | 1.77 | 0.68 | 1.45 |
| v_70000 | **0.60** | **0.90** | 0.81 | 6.52 | 0.80 | 1.60 | 0.69 | 1.40 |
| v_85000 | **0.60** | **0.91** | 0.80 | 4.78 | 0.80 | 1.60 | 0.74 | 1.43 |
| v_92500 | **0.57** | **0.88** | 1.36 | 4.80 | 0.76 | 1.52 | 0.74 | 1.47 |
| v_100000 | **0.46** | **0.62** | 4.04 | 3.34 | 0.60 | 1.01 | 0.65 | 1.08 |
| t_5000 | 0.32 | **0.35** | 0.43 | 0.43 | 0.51 | 0.63 | **0.29** | 0.63 |
| t_10000 | **0.24** | **0.29** | 0.35 | 0.39 | 0.41 | 0.90 | 0.25 | 0.57 |
| t_15000 | **0.23** | **0.29** | 0.33 | 0.36 | 0.36 | 0.82 | 0.26 | 0.49 |
| t_20000 | **0.31** | **0.34** | 0.41 | 0.46 | 0.43 | 0.63 | 0.35 | 0.43 |
| t_25000 | **0.25** | **0.28** | 0.34 | 5.42 | 0.38 | 0.83 | 0.29 | 0.38 |
| t_30000 | **0.21** | **0.26** | 0.28 | 1.29 | 0.34 | 0.92 | 0.25 | 0.39 |
| t_40000 | **0.21** | **0.28** | 0.28 | 1.57 | 0.35 | 0.96 | 0.25 | 0.43 |
| t_50000 | **0.19** | **0.27** | 0.31 | 1.43 | 0.35 | 0.89 | 0.24 | 0.44 |
| t_60000 | **0.18** | **0.26** | 0.28 | 1.47 | 0.33 | 0.87 | 0.22 | 0.43 |
| t_70000 | **0.21** | **0.30** | 0.30 | 7.29 | 0.38 | 0.98 | 0.26 | 0.51 |
| t_85000 | **0.28** | **0.35** | 0.44 | 2.64 | 0.49 | 1.15 | 0.38 | 0.65 |
| t_92500 | **0.31** | **0.37** | 1.05 | 3.02 | 0.53 | 1.20 | 0.43 | 0.75 |
| t_100000 | **0.29** | **0.36** | 9.79 | 3.11 | 0.52 | 1.19 | 0.48 | 0.95 |
| q_5000 | 6.30e-08 | 5.61e-08 | 7.99e-08 | 6.99e-08 | 8.89e-08 | 1.49e-07 | **5.03e-08** | **5.51e-08** |
| q_10000 | **1.87e-07** | 3.86e-07 | 2.42e-07 | **3.66e-07** | 4.20e-07 | 6.99e-07 | 4.45e-07 | 6.63e-07 |
| q_15000 | **6.13e-07** | **1.17e-06** | 7.38e-07 | 1.18e-06 | 8.85e-07 | 1.66e-06 | 9.66e-07 | 2.13e-06 |
| q_20000 | **3.02e-06** | **5.68e-06** | 3.86e-06 | 7.74e-06 | 3.94e-06 | 8.34e-06 | 4.73e-06 | 1.05e-05 |
| q_25000 | **9.16e-06** | **1.72e-05** | 1.17e-05 | 9.99e-05 | 1.21e-05 | 2.72e-05 | 1.47e-05 | 3.31e-05 |
| q_30000 | **2.03e-05** | **3.85e-05** | 2.58e-05 | 1.02e-04 | 2.70e-05 | 6.43e-05 | 3.17e-05 | 7.52e-05 |
| q_40000 | **5.82e-05** | **1.13e-04** | 7.41e-05 | 3.92e-04 | 7.71e-05 | 1.93e-04 | 8.62e-05 | 2.15e-04 |
| q_50000 | **1.13e-04** | **2.16e-04** | 1.53e-04 | 7.44e-04 | 1.46e-04 | 3.74e-04 | 1.56e-04 | 3.72e-04 |
| q_60000 | **1.79e-04** | **3.34e-04** | 2.23e-04 | 1.05e-03 | 2.26e-04 | 5.60e-04 | 2.32e-04 | 5.10e-04 |
| q_70000 | **2.56e-04** | **4.76e-04** | 3.14e-04 | 3.60e-03 | 3.20e-04 | 7.35e-04 | 3.22e-04 | 6.76e-04 |
| q_85000 | **4.13e-04** | **6.60e-04** | 5.04e-04 | 3.27e-03 | 5.02e-04 | 9.85e-04 | 4.83e-04 | 9.04e-04 |
| q_92500 | **4.57e-04** | **5.77e-04** | 1.28e-03 | 3.37e-03 | 5.78e-04 | 1.03e-03 | 5.44e-04 | 8.06e-04 |
| q_100000 | **3.59e-04** | **4.32e-04** | 5.13e-03 | 2.44e-03 | 4.97e-04 | 9.81e-04 | 4.99e-04 | 8.01e-04 |

Table 5: Losses of each variable for each model, follow Table 4

| Variable | Definition |
|---|---|
| Debris Flow | May be triggered by intense rain after wildfires. A slurry of loose soil, rock, organic matter, and water, similar to wet concrete, which is capable of holding particles larger than gravel in suspension. |
| Flash Flood | A life-threatening, rapid rise of water into a normally dry area beginning within minutes to multiple hours of the causative event (e.g., intense rainfall, dam failure, ice jam). |
| Flood | Any high flow, overflow, or inundation by water which causes damage. In general, this would mean the inundation of a normally dry area caused by an increased water level in an established watercourse, or ponding of water, associated with heavy rainfall |
| Funnel Cloud | A rotating, visible extension of a cloud pendant from a convective cloud with circulation not reaching the ground. It is a precursor to more severe weather events. The wind shear caused by it (sudden changes in wind speed and direction) will endanger aviation. |
| Hail | Frozen precipitation in the form of balls or irregular lumps of ice. |
| Heavy Rain | Unusually large amount of rain which does not cause a Flash Flood or Flood event, but causes damage |
| Lightning | A sudden electrical discharge from a thunderstorm, resulting in a fatality, injury, and/or damage |
| Marine Hail | Hail 3/4 of an inch in diameter or larger, occurring over the waters and bays of the ocean, Great Lakes, and other lakes with assigned specific Marine Forecast Zones |
| Marine High Wind | Non-convective, sustained winds or frequent gusts of 48 knots (55 mph) or more, resulting in a fatality, injury, or damage, over the waters and bays of the ocean, Great Lakes, and other lakes with assigned specific Marine Forecast Zones. |
| Marine Strong Wind | Non-convective, sustained winds or frequent gusts up to 47 knots (54 mph), resulting in a fatality, injury, or damage, occurring over the waters and bays of the ocean, Great Lakes, and other lakes with assigned specific Marine Forecast Zones. |
| Marine Thunderstorm Wind | Winds, associated with thunderstorms, occurring over the waters and bays of the ocean, Great Lakes, and other lakes with assigned specific Marine Forecast Zones with speeds of at least 34 knots (39 mph) for 2 hours or less. |
| Thunderstorm Wind | Winds, arising from convection (occurring within 30 minutes of lightning being observed or detected), with speeds of at least 50 knots (58 mph), or winds of any speed (non-severe thunderstorm winds below 50 knots) |
| Tornado | A violently rotating column of air, extending to or from a cumuliform cloud or underneath a cumuliform cloud, to the ground, and often (but not always) visible as a condensation funnel. |
| Waterspout | A rotating column of air, pendant from a convective cloud, with its circulation extending from cloud base to the water surface of bays and waters of the Great Lakes, and other lakes with assigned Marine Forecast Zones. |
| Wind | Severe thunderstorm wind, strong wind that causes damage, which is reported and recorded by NOAA Storm Prediction Center (National Oceanic and Atmospheric Administration (NOAA), Storm Prediction Center (SPC), 2024). |
| Heat | Large area of excessive heat above 37 degrees celsius in US during daytime, manually filtered. |
| Cold | Large area of excessive cold below -29 degress celsius in US during nighttime, manually filtered. |

Table 6: Explaination of each type of extreme weather included in HR-Extreme

