# HR-Extreme: Supplementary Materials

**Nian Ran[1], Peng Xiao[2], Yue Wang[1]\*, Wenlei Shi[3], Jiaxin Lin[2], Qi Meng[4], Richard Allmendinger[5]\***
[1]Zhongguancun Academy
[2]Hunan University
[3]Microsoft Research
[4]Chinese Academy of Sciences
[5]University of Manchester

## CONTENTS

---

\*Corresponding author,(yuewang@bjzgca.edu.cn, richard.allmendinger@manchester.ac.uk)

# 1 DATASET AND CODE ACCESS

Following the instructions of NeurIPS Dataset and Benchmark Track guidelines, we have uploaded our datasets to Hugging Face and code for constructing dataset to Github:

1. HR-Extreme: `https://huggingface.co/datasets/NianRan1/HR-Extreme`

2. HR-Extreme Croissant metadata: `https://huggingface.co/api/datasets/NianRan1/HR-Extreme/croissant`

3. Code: `https://github.com/HuskyNian/HR-Extreme/tree/main`

The access to NOAA Storm Event Database is `https://www.ncdc.noaa.gov/stormevents/ftp.jsp` and NOAA Storm Prediction Center is `https://www.spc.noaa.gov/climo/reports/`. Dataset and code will be available upon the acceptance of this paper.

## 1.1 VARIABLE LIST

There are 69 physical variables in total included in HR-Extreme, details are explained in Table 1. The description of each type of event included is in Table 4.

| Variable | Definition | Unit | Range |
|----------|------------|------|-------|
| msl | Mean Sea Level Pressure | Pa | |
| 2t | Temperature 2 m above ground | K | |
| 10u | U-component Wind Speed 10 m above ground | m/s | |
| 10v | V-component Wind Speed 10 m above ground | m/s | |
| hgtn | Geopotential Height | gpm | |
| u | U-component Wind Speed | m/s | At 50, 100, 150, 200, 250, 300, |
| v | V-component Wind Speed | m/s | 400, 500, 600, 700, 850, 925, 1000 |
| t | Temperature | K | millibars, 13 levels in total |
| q | Specific Humidity | kg/kg | |

Table 1: Summary of the 69 physical variables in the dataset

## 1.2 DATASET STATISTICS

We calculated the means and standard deviations of the HRRR (National Oceanic and Atmospheric Administration (NOAA), 2024) data to normalize the data before inputting it into deep learning models. These statistics are stored in a file accessible at `https://huggingface.co/datasets/NianRan1/HR-Extreme/tree/main/statistics`. Our dataset repository is organized into two distinct directories, *202001_202006* and *202007_202012*, which contain data from January 2020 to June 2020 and from July 2020 to December 2020, respectively. Within each directory, the dataset is stored in the WebDataset [1] format, adhering to Hugging Face's guidelines. Specifically, every 10 *npz* files are aggregated into a single *tar* file, named sequentially as *"i.tar"*, where *i* is an integer (e.g., *"0001.tar"*). Consequently, a total of 2200 files will be organized into 220 *tar* files.

## 1.3 CODE GUIDANCE

We provide our code on GitHub for constructing the dataset, considering different usages and potential future developments. The repository includes three Python scripts named *make_dataseti.py"*, where *i* is 1, 2, or 3. These scripts correspond to three data sources: events from the Storm Events

---

[1] `https://github.com/webdataset/webdataset`

Database (National Centers for Environmental Information (NCEI), 2024), events from the Storm Prediction Center (National Oceanic and Atmospheric Administration (NOAA), Storm Prediction Center (SPC), 2024), and excessive heat and cold events filtered manually, respectively. Users, however, need only utilize *"make_datasetall.py"*, supplying start and end dates to generate a comprehensive index file. This index file delineates the events, including their types, bounding box indices, event start and end times, and details of event spans if multiple types are involved. For instance, executing *"python make_datasetall.py 20200101 20200630"* will produce an index file named *"data_20200101_20200630_info.csv"* within the *"index_files"* directory, ready for generating bounding boxes from HRRR data.

It is important to note that users must first download a storm event details CSV file from the Storm Events Database [2], specifically the *StormEvents_details-ftp_v1.0_year_cdate.csv.gz"* file, and place it in the *"index_files"* directory before executing any of the Python scripts. Users should ensure they download the storm event details files, rather than location or other types of files, and confirm that these files cover the dates of the dataset they intend to create.

### 1.4 TARGET AUDIENCE

HR-Extreme is a dataset encompassing 17 types of extreme weather events for the year 2020, derived from HRRR (National Oceanic and Atmospheric Administration (NOAA), 2024) data. Users can conveniently extend this dataset using the provided code interface. The dataset is intended for a wide range of researchers in the field of weather forecasting, including those utilizing physical methods and deep learning techniques. Given the critical importance of accurately predicting extreme weather events in weather forecasting, it is imperative that researchers evaluate their models using this dataset. This evaluation will help to identify the strengths and weaknesses of their models and assess their practical applicability. While most previous studies in deep learning have focused on ERA5 (Hersbach et al., 2020) data, models can be readily adapted to work with HRRR data as well.

### 1.5 LICENSES

The license of NOAA HRRR data is U.S Government Work [3], which means they can be used by anyone for any purpose without seeking permission or paying royalties.

### 1.6 AUTHOR STATEMENT

As the authors of the HR-Extreme dataset, we bear full responsibility for any violations of rights and confirm that the dataset is released under the appropriate data license. The data has been compiled and shared in accordance with the relevant guidelines and regulations.

## 2 EXPERIMENT MODELS

### 2.1 NWP

The Numerical Weather Prediction (NWP) (Olson et al., 2022) system for predicting High-Resolution Rapid Refresh (HRRR) data is a sophisticated model that leverages high-performance computing to simulate atmospheric conditions with fine temporal and spatial resolution. HRRR is a real-time 3-km resolution, hourly updated, cloud-resolving, convection-allowing atmospheric model, initialized by 3-km grids with 3-km radar assimilation. It is designed to provide detailed forecasts over the contiguous United States for short-term predictions up to 18 hours. The HRRR system integrates a variety of data sources, including radar, satellite, and surface observations, to continuously update its predictions. It employs advanced physics parameterizations to accurately represent processes such as convection, cloud formation, and precipitation. By using a high-resolution grid, HRRR is able to capture small-scale weather phenomena, such as thunderstorms and tornadoes, with greater accuracy. The model's outputs are crucial for various applications, including aviation, re-

---

[2] https://www.ncei.noaa.gov/pub/data/swdi/stormevents/csvfiles/
[3] https://registry.opendata.aws/noaa-hrrr-pds/

newable energy, and severe weather forecasting, providing timely and precise information to support decision-making and public safety.

## 2.2 PANGU

The Pangu-Weather (Bi et al., 2023) model is an deep learning-based weather forecasting system that employs a three-dimensional deep neural network architecture designed specifically for Earth's unique geospatial features. The core of Pangu-Weather is the 3D Earth-specific transformer (3DEST), which incorporates height information as an additional dimension, allowing the model to effectively capture atmospheric states across various pressure levels. This architecture uses an encoder-decoder structure derived from the Swin transformer (Liu et al., 2021), with enhancements such as Earth-specific positional biases to handle geospatial relationships more accurately.

To make predictions, Pangu-Weather takes reanalysis weather data as input and processes it through a hierarchical temporal aggregation strategy. This strategy trains multiple models with varying lead times (1 hour, 3 hours, 6 hours, and 24 hours) and uses a greedy algorithm during inference to minimize the number of iterations required for medium-range forecasts. By iteratively using forecast results as new inputs, the model reduces cumulative errors and improves accuracy. The combination of 3D spatial awareness and hierarchical temporal aggregation enables Pangu-Weather to produce faster and more accurate weather forecasts, outperforming traditional numerical weather prediction systems like the ECMWF's IFS in various metrics including root mean square error (RMSE) and anomaly correlation coefficient (ACC).

## 2.3 FUXI

The Fuxi (Chen et al., 2023) model is another deep learning-based weather forecasting system that employs a cascaded architecture to generate 15-day global forecasts with a temporal resolution of 6 hours and a spatial resolution of $0.25°$. This model is built on the U-Transformer architecture, which effectively captures complex spatiotemporal relationships in high-dimensional weather data. The system uses a pre-trained base model, which is fine-tuned for specific forecast time windows: 0-5 days, 5-10 days, and 10-15 days. By cascading these models, Fuxi reduces error accumulation and enhances forecast accuracy over longer periods. The model was trained on 39 years of ECMWF ERA5 (Hersbach et al., 2020) reanalysis data. During inference, Fuxi utilizes an autoregressive approach where the outputs of one model serve as inputs for the next, significantly extending the skillful forecast lead time. Additionally, Fuxi employs ensemble forecasting by perturbing initial conditions and model parameters, providing a measure of forecast uncertainty and demonstrating performance comparable to the ECMWF ensemble mean for the 15-day forecast horizon.

## 2.4 HR-HEIM

**HR-Heim** The architecture of HR-Heim follows a conventional structure with an encoder, a series of transformer layers, and a decoder, inspired by the FuXi architecture (Chen et al., 2023). For the encoder, we utilize causal convolutions from MagViTv2 to capture spatial-temporal input (Yu et al., 2023). The transformer segment consists of multiple stacked SwinTransformer blocks (Liu et al., 2022). Unlike typical Vision Transformer decoders that use a simple MLP with $1 \times 1$ convolution, which can hinder resolution, our decoder progressively upscales the feature map from $\frac{H}{h} \times \frac{W}{w}$ to the target size $H \times W$ through a series of steps. Each step resolves details at its specific resolution level, incorporating convolutional layers and upsampling operations to enhance prediction quality.

**Decoder Grouping by Variable Types** In the context of a multi-task formulation, the optimal approach for performance would typically involve training a separate model for each subtask. However, this becomes inefficient and scales poorly during training and inference when dealing with hundreds of physical variables. To address this, we propose a shared backbone for feature extraction and temporal modeling, which we assume to be universally applicable for predicting any physical variables. We then separate multiple decoding heads for different variables.

To further enhance efficiency, we categorize the subtasks into several groups, assigning a decoding head to each. The grouping principle aims to minimize negative transfer within each group. For example, during training, the gradients of subtask A and subtask B often point in conflicting directions.

Following previous work, we train a simple model that shares all parameters and assesses conflicts and alignment between each pair of subtasks during training. Specifically, we compute the cosine similarity of the gradients of each pair of subtasks to obtain an affinity matrix.

$$\hat{Z}_{ij} = \frac{1}{T} \sum_{t=1}^{T} \frac{g_t^i \cdot g_t^j}{\left\| g_t^i \right\|_2 \left\| g_t^j \right\|_2} \tag{1}$$

Here, $g_t^i$ represents the flattened gradient from the loss of the $i$th subtask. Interestingly, we find that variables of the same type but at consecutive pressure levels (e.g., $t\_100$ and $t\_120$) align better than those of the same pressure level but different types (e.g., $t_1 00$ and $h_1 00$). Consequently, each group contains only one type of variable, and we determine which pressure levels to group together by maximizing the inner group affinity.

## 3 MODEL EVALUATIONS

### 3.1 METRICS

The Root Mean Square Error (RMSE) is a commonly used metric to measure the difference between values predicted by a model and the values actually observed. It is defined as the square root of the mean of the squared differences between the predicted and observed values.

Mathematically, RMSE is expressed as:

$$\text{RMSE} = \sqrt{\frac{1}{n} \sum_{i=1}^{n} (y_i - \hat{y}_i)^2}$$

where $n$ is the number of observations, $y_i$ is the observed value, $\hat{y}_i$ is the predicted value. The RMSE value provides a measure of how well a model predicts the outcome variable. Lower RMSE values indicate better fit, meaning the predicted values are closer to the observed values.

### 3.2 RESULTS

Our full results are included in Table 2 and Table 3, showing the RMSE for each variable predicted by each model.

## 4 DATASHEET

### 4.1 MOTIVATION

1. **For what purpose was the dataset created?** *We propose HR-Extreme, a dataset with high-resolution feature maps of physical variables for evaluating cutting-edge models in extreme weather prediction. This crucial aspect has been overlooked and undervalued in previous work, and there is a lack of dedicated high-resolution datasets for extreme weather forecasting.*

2. **Who created the dataset and on behalf of which entity?** *The dataset was developed by a consortium of ML researchers and climate scientists listed in the author list.*

3. **Who funded the creation of the dataset?** *Self-funded.*

### 4.2 DISTRIBUTION

1. **Will the dataset be distributed to third parties outside of the entity (e.g., company, institution, organization) on behalf of which the dataset was created?** *Yes, it is open the public.*

2. **How will the dataset will be distributed (e.g., tarball on website, API, GitHub)?** *The dataset will be distributed in Hugging Face, and code for constructing the dataset held in Github.*

3. **Have any third parties imposed IP-based or other restrictions on the data associated with the instances?** *No.*

4. **Do any export controls or other regulatory restrictions apply to the dataset or to individual instances?** *No.*

### 4.3 MAINTENANCE

1. **Who will be supporting/hosting/maintaining the dataset?** *The authors of this paper.*

2. **How can the owner/curator/manager of the dataset be contacted(e.g., email address)?** *The owner/curator/manager of the dataset can be contacted by Nian Ran (r992988188@gmail.com).*

3. **Is there an erratum** *No. If errors are found in the future, we will release errata on the Github page:* `https://github.com/HuskyNian/HR-Extreme/tree/main`.

4. **Will the dataset be updated (e.g., to correct labeling errors, add new instances, delete instances)?** *Yes, we will update our dataset whenever necessary to ensure accuracy, and announcements will be made accordingly. The updates will be shown in our Github page:* `https://github.com/HuskyNian/HR-Extreme/tree/main`.

5. **If the dataset relates to people, are there applicable limits on the retention of the data associated with the instances (e.g., were the individuals in question told that their data would be retained for a fixed period of time and then deleted?)** *N/A*

6. **Will older version of the dataset continue to supported/hosted/maintained?** *Yes, they will be still available and easy to reproduce by our provided code.*

7. **If others want to extend/augment/build on/contribute to the dataset, is there a mechanisms for them to do so?** *Yes, they can use our provided code on (*`https://github.com/HuskyNian/HR-Extreme/tree/main`*), we have provided full instructions in our paper and this material.*

### 4.4 COMPOSITION

1. **What do the instance that comprise the dataset represent (e.g., documents, photos, people, countries?)** *Each instance is the feature maps with 320 in width and 320 in height, and 69 channels, which representing 69 physical variables as shown in Table 1. This areas are cropped from HRRR (National Oceanic and Atmospheric Administration (NOAA), 2024) data.*

2. **How many instances are there in total (of each type, if appropriate)?** *There are 22,774 files accounting for the period of July 2020 to December 2020, and 34,196 files for the period of January 2020 to June 2020.*

3. **Does the dataset contain all possible instances or is it a sample of instances from a larger set?** *Our dataset contains all possible instances in entire 2020. Users can use our code to extend to more years depending on their needs.*

4. **Is there a label or target associated with each instance?** *Yes, after loading a file by **Numpy** library, users can use keys "inputs" to retrieve inputs and "targets" to retrieve targets and "mask" to retrieve mask for that case.*

5. **. Is any information missing from individual instances?** *No.*

6. **Are there recommended data splits (e.g., training, development/validation, testing)?** *Considering the computational and memory requirements, we recommend using the data from July 2020 to December 2020 for evaluation. Our evaluation results are based on this subset.*

7. **Are there any errors, sources of noise, or redundancies in the dataset?** *Because our dataset is primarily based on user reports, there may be noise and errors in the range and span of each extreme event case. To mitigate these errors, we have increased the range of each case.*

8. **Is the dataset self-contained, or does it link to or otherwise rely on external resources (e.g., websites, tweets, other datasets)?** *The dataset is self-contained.*

9. **Does the dataset contain data that might be considered confidential?** *No.*

10. **Does the dataset contain data that, if viewed directly, might be offensive, insulting, threatening, or might otherwise cause anxiety?** *No.*

### 4.5 COLLECTION PROCESS

1. **How was the data associated with each instance acquired?** *Each instance is cropped from HRRR (National Oceanic and Atmospheric Administration (NOAA), 2024) by a prepared index file, the index file is constructed by data from NOAA Storm Event Database (`https://www.ncdc.noaa.gov/stormevents/ftp.jsp`) and NOAA Storm Prediction Center (`https://www.spc.noaa.gov/climo/reports/`).*

2. **What mechanisms or procedures were used to collect the data (e.g., hardware apparatus or sensor, manual human curation, software program, software API)?** *We use many CPU nodes to to process the data with Python libraries to access HRRR data. Users are recommended to first download ground true HRRR data in a storage for easy and frequent access.*

3. **Who was involved in the data collection process (e.g., students, crowdworkers, contractors) and how were they compensated (e.g., how much were crowdworkers paid)?** *Regular students and employees are involved, no crowdworkers are involved.*

4. **Does the dataset relate to people?** *No.*

5. **Did you collect the data from the individuals in questions directly, or obtain it via third parties or other sources (e.g., websites)?** *We obtained the dataset from open resource HRRR (National Oceanic and Atmospheric Administration (NOAA), 2024) data.*

### 4.6 USES

1. **Has the dataset been used for any tasks already?** *No, this dataset has only been evaluated on some models explained in the paper.*

2. **What (other) tasks could be the dataset be used for?** *Apart from evaluation, it can be used a finetune dataset for deep learning models.*

3. **Is there anything about the composition of the dataset or the way it was collected and preprocessed/cleaned/labeled that might impact future uses?** *The varibale choices can impact the future use. But so far the variables chosen are diverse and sufficient for evaluating extreme weather.*

4. **Are there tasks for which the dataset should not be used?** *No.*

## 5 ADDITIONAL FIGURES AND TABLES

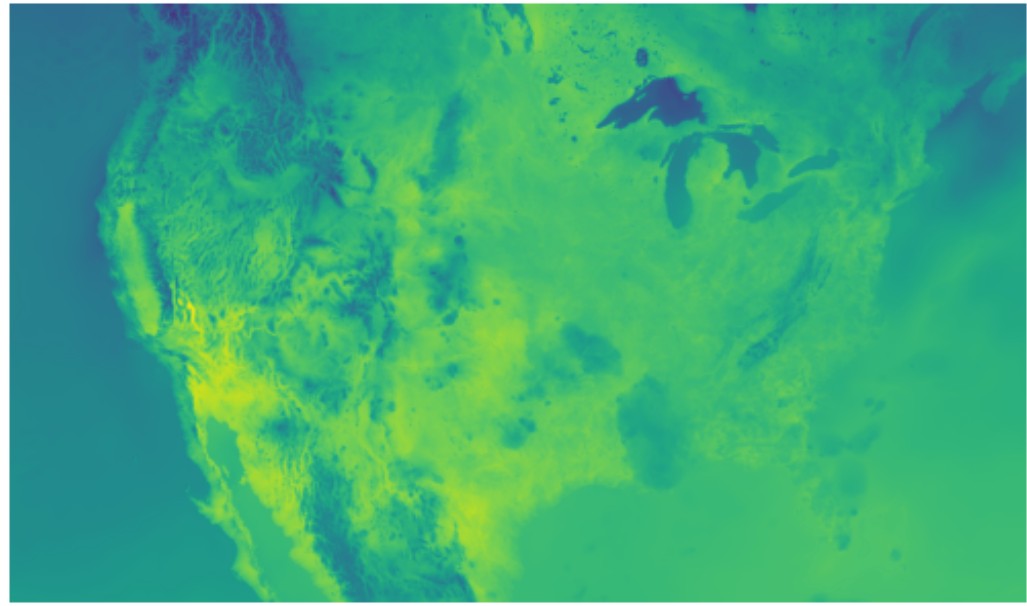

Figure 1: An example plot of the variable "temeprature 2 meters above ground" of HRRR data on U.S.

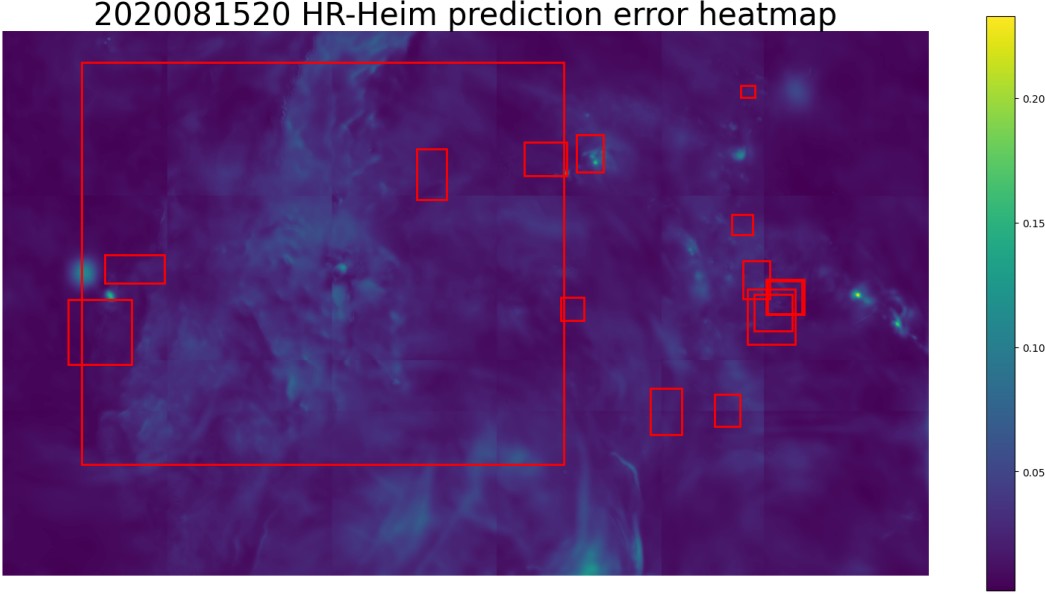

Figure 2: More examples: the mean error heatmap of all variables predicted by HR-Heim on entire U.S. on 8 p.m., 15 August. Bright area means larger avaergae loss. They are captured by the bounding boxes representing the extreme weather that is happening.

## REFERENCES

Kaifeng Bi, Lingxi Xie, Hengheng Zhang, Xin Chen, Xiaotao Gu, and Qi Tian. Accurate medium-range global weather forecasting with 3d neural networks. *Nature*, 619(7970):533–538, July 2023. ISSN 1476-4687. doi: 10.1038/s41586-023-06185-3. URL https://doi.org/10.1038/s41586-023-06185-3.

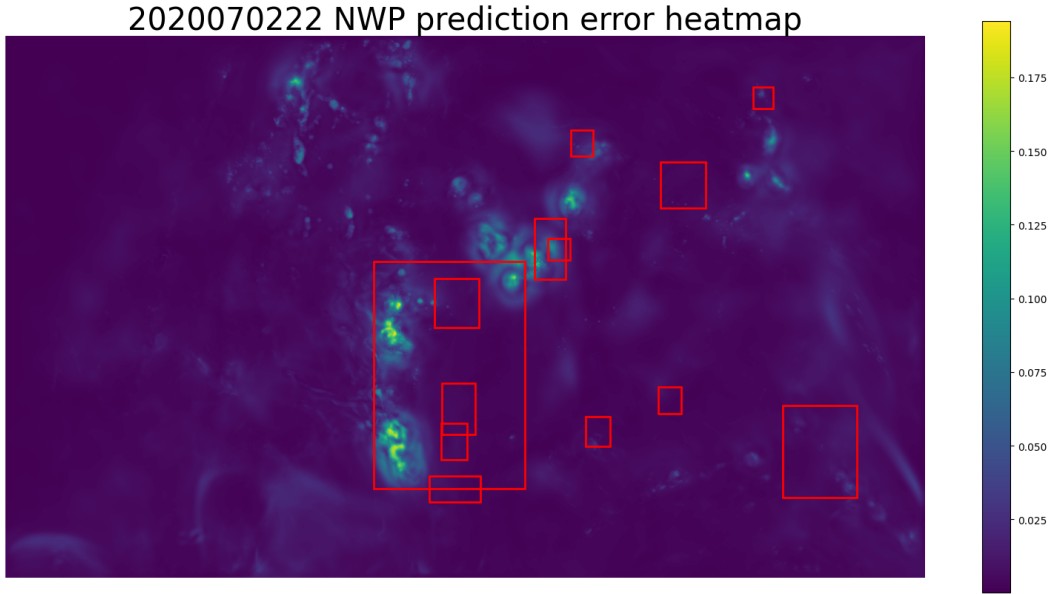

Figure 3: More examples: the mean error heatmap of all variables predicted by NWP on entire U.S. on 10 p.m., 2 July. Bright area means larger avaergae loss. They are captured by the bounding boxes representing the extreme weather that is happening.