# OpenReview forum: "HR-Extreme: A High-Resolution Dataset for Extreme Weather Forecasting"
_ICLR.cc/2025/Conference — ICLR 2025 Poster_

### Official Review · Reviewer_tGGo · 2024-10-27

**Soundness:** 2
**Presentation:** 2
**Contribution:** 3
**Rating:** 8
**Confidence:** 4

**Summary:**

This paper introduces a high-resolution dataset, called HR-Extreme, for numerical weather forecasting under extreme weather conditions, an area often overlooked in weather forecasting literature. The authors also present a baseline deep learning model alongside the dataset, called HR-Heim, which outperforms state-of-the-art weather models in extreme weather conditions.

**Strengths:**

* The authors provide a high-resolution dataset for numerical weather forecasting under extreme weather conditions. The need for such dataset has been evident for some time, as highlighted by the low performance analysis of SOTA weather forecasting models.
* The construction of the dataset is well-motivated, and the authors provide a clear and thorough explanation of both the data collection and construction processes.
* Additionally, the authors provide a baseline deep learning model called HR-Heim, which is inspired by a SOTA numerical weather forecasting model, to specifically excel and outperform SOTA models under extreme weather conditions.

**Weaknesses:**

* In Section 2.2, the authors discuss some datasets that share similarities (to a certain extent) with the proposed one. It would have been beneficial to illustrate the performance of the SOTA models and the proposed HR-Heim on some of these datasets. Specifically, the last dataset introduced by Liu et al., which also includes certain extreme weather conditions, could have been included in the experiments to support the need for HR-Extreme and support the performance of HR-Heim. Given that HR-Heim outperforms SOTA methods in HR-Extreme, we would expect to see similar results on other datasets with extreme weather conditions.
* As the authors also state as a limitation, this kind of study should include more than just a single-step prediction analysis. Given the increased difficulty of predicting extreme weather events in the long term, an accuracy comparison between HR-Heim and SOTA methods across varying time horizons could be valuable.

**Questions:**

* Could you extend the dataset analysis to include predictions for more than a single hour ahead?
* As an extension to the previous question, could you analyze and compare the performance of state-of-the-art models and HR-Heim for both short-term and long-term forecasting?
* Could you further evaluate the performance of HR-Heim on a similar extreme weather dataset?

---

> ### Author Response · Authors · 2024-11-22
> **Response to Reviewer tGGo [Part 1]**
>
> Thank you for your positive feedback. We truly appreciate your insightful understanding and recognition of the key contributions of our study in terms of significance, comprehensiveness and a strong baseline model for weather prediction on HRRR data! We have revised our paper according to your suggestions and provide our responses to address your concerns below.
>
> > **More experiments with SOTA models and HR-Heim. Include prediction for more than a single hour ahead**
>
> **A**: Thank you for your suggestions. We have finished the experiment of NWP on HRRR original data, NWP model and HR-Heim on HR-Extreme for lead time from 1 to 4 and added to the end of Section 4.3 in blue. There is also additional statistical analysis in Appendix A.1. The experiments shows HR-Heim consistently outperform others in all lead times. Training and evaluating is highly resource-intensive and time-consuming, however, we want to note that the primary goal of this work is to address the gap in high-resolution datasets for extreme weather evaluation, encompassing comprehensive event types rather than HR-Heim model. We aim to highlight the significance of extreme weather prediction for future research, demonstrate the substantial room for improvement in the performance of current SOTA models. Then additionally, we present a model improved for high-resolution prediction, which achieves better results on HRRR data and our extreme weather dataset. This serves as strong a baseline model for future predictions on HRRR data and its extreme events.
>
> > **Evaluate HR-Heim on a similar extreme weather dataset**
>
> **A**: Thank you for your suggestion. The experiment of HR-Heim at different lead times are added to paper at the end of Section 4.3 in blue, showing its consistent superior performance. HRRR data offers significantly higher resolution than ERA5, while the dataset you referenced is built on ERA5. In contrast, HR-Heim is specifically optimized for high-resolution prediction on HRRR and trained on HRRR data. Nonetheless, our dataset addresses the gap in high-resolution datasets for extreme weather, with its primary contribution being its application for model evaluation.

---

> > ### Author Response · Authors · 2024-11-29
> > **Response**
> >
> > Thank you once again for your valuable comments and suggestions, which have greatly improved our paper. We have carefully addressed your concerns by revising several sections and conducting additional experiments and analyses. Could you kindly let us know if you have any further concerns or suggestions?

---

> > ### Comment · Reviewer_tGGo · 2024-11-30
> > **Response to Authors**
> >
> > The updated experiments involving HR-Heim seem to answer my concerns. I am updating my score accordingly.

---

> ### Author Response · Authors · 2024-12-01
> **Thanks!**
>
> Thank you for taking the time to reassess your concerns regarding our work, we are grateful for your constructive engagement!

---

### Official Review · Reviewer_kLu6 · 2024-11-02

**Soundness:** 3
**Presentation:** 3
**Contribution:** 2
**Rating:** 6
**Confidence:** 4

**Summary:**

The authors present a new dataset of labelled extreme weather events over the continental US based on a high resolution (3km) numerical forecast product. They compare the events from a numerical weather prediction (NWP) model as well as two baseline ML based weather models and a newly proposed variant. The authors claim improved skill in their new variant compared to the other baselines.

**Strengths:**

The constructed dataset could provide a useful evaluation of extreme events in new ML based weather prediction models which tend to be evaluated on larger scale statistics, potentially hiding biases in such events. It is based on an operational high-resolution dataset with a mixture of automated and manual labelling.

**Weaknesses:**

There are a number of technical and fundamental weaknesses that undermine the above strengths.

Major issues:
1) Extreme events are, by definition, rare. While a database of such events derived from a high-resolution reanalysis product could provide a useful starting point for evaluation, the specific events are much less useful without a probabilistic understanding or measure of their likelihood at different lead times. i.e. It is probably more likely that any of the given events *wouldn't* have happened given a similar atmospheric state if the conditions were encountered again. Without a probabilistic understanding of the probability of an event (based on lots of comparable events that weren't extreme), this dataset has very little value as presented.
 2) This leads to my second concern regarding the evaluation setup. Given the dataset extracts the atmospheric state at t=0, -1 and -2 hours, I presume the evaluations happen from an atmospheric state of t=-2 hours and run forward? This is no longer forecasting, but nowcasting and quite a different task. Since the atmospheric state already has the extreme event and the model just needs to advect it correctly. I have no idea how the NWP model is compared in this setting since presumably the authors don't run this explicitly with the extracted state? Or perhaps they do? Also, since the authors use the same NWP as was used to create the HRRR dataset, why doesn't it perform as well as the other models? At what resolution are the models run, are they run globally, over the US, or only over the event region? Are the global Fuxi and Pangu models retrained for these regions?
 3) Related to this, to what extent is this dataset used to train the different models? Does HR-Helm get to see some or any extreme event data during training? Given the issues described in (1), how do you avoid overfitting?

One more minor issue is that I would like to see Figure 4 presented for the same day for all 4 models, with a separate figure for the other day for all 4 models in the appendix. Currently it's impossible to fairly compare the skill of the models (although it seems like the NWP already does better than HR-Helm across the two examples).

**Questions:**

Please provide specific responses to each of the concerns and questions raised above in the weaknesses.

---

> ### Author Response · Authors · 2024-11-22
> **Response to Reviewer kLu6 [Part 1]**
>
> Thank you for your review and we understand your feedback. It seems there may have been some misunderstanding regarding the usage and purpose of our dataset, as well as the settings of the HRRR data.
>
> > **Concerns about the significance of our work**
>
> **A**:
> - **Clarification of Dataset Purpose**: We understand that you may view our work as a dataset for training models to better predict extreme events. However, our work is actually a dataset for evaluating the performance of models on recorded extreme events based on HRRR data. We understand your concern about the uncertainty of extreme weather and may think that our extreme dataset is built on a simulator. However, our dataset is systematically constructed from recorded extreme events and is primarily intended for evaluation. We have also stated in the future work in our paper that it is only a possibility for future models to fine-tune on our dataset to improve their performance.
>
> - **Contributions and Significance**: The primary goal of this work is to address the gap in high-resolution datasets for extreme weather evaluation, encompassing comprehensive event types. We provide such a dataset to evaluate the ability of SOTA models on recorded extreme events. We aim to highlight the significance of extreme weather prediction for future research, demonstrate the substantial room for improvement in the performance of current SOTA models. Additionally, we present a model improved for high-resolution prediction, which achieves better results on HRRR data and our extreme weather dataset. This serves as a baseline model for future predictions on HRRR data and its extreme events.
>
> - **Correctness of our Dataset and NWP Model Usage**: You might have been misled by the HRRR dataset settings. The analysis at lead time 0 (often referred to as f00) is a blend of the model's background (a short-term forecast from the previous cycle) and recent observations, integrated through data assimilation techniques. This process incorporates various observational data, such as from aircraft, radar ..., to produce an accurate representation of the current atmospheric state. Therefore, while the f00 output is heavily informed by actual observations. For lead times greater than 0 (e.g., f01, f02), the outputs are forecasts generated by the model based on the initial analysis [3]. And our dataset is built on data at lead time 0, the predictions of NWP uses lead time at least 1, which enables fair comparison to other models.
>
> - **Importance of Extreme Weather Nowcasting**: We want to state that extreme weather nowcasting is also a crucial task in practice, such as work NowcastNet[1] and DGMR[2] published on Nature. We have already considered the unusual atmosphere states before and after the extreme events, therefore we incorporate several hours of atmospheric state data before and after the recorded events to include the comprehensive progression of an extreme event. On top of that, our open-source code is designed to allow users to adjust the temporal dimension, enabling arbitrary lead-time predictions or extending the atmospheric state analysis before and after the event. This flexibility supports both nowcasting and longer forecasting evaluation.
>
> >
> [1] Zhang, Y., Long, M., Chen, K. et al. Skilful nowcasting of extreme precipitation with NowcastNet. Nature 619, 526–532 (2023).
> [2]Ravuri, S., Lenc, K., Willson, M. et al. Skilful precipitation nowcasting using deep generative models of radar. Nature 597, 672–677 (2021)
>
> [3] How do I download 15-minute HRRR data: https://github.com/blaylockbk/Herbie/discussions/67#discussioncomment-2537530
>
> > **Question about model training**
>
> **A**: Thank you for your question. All models (Pangu, Fuxi, and our HR-Heim) were trained on HRRR data spanning the U.S. from January 2019 to June 2020, from scratch. They were trained under identical parameters and same level of model parameters, and no hyperparameter tuning was applied to HR-Heim. Furthermore, none of these models were fine-tuned on our HR-Extreme dataset, ensuring a fair basis for comparison and evaluation, which means none of the models have seen the extreme event in training. This clarification has been added at the beginning of Section 4.3
>
> > **Concerns about Figure 4**
>
> **A**:  The intention is not to compare the models’ abilities to predict extreme events but to validate the dataset’s capability in identifying high-error regions associated with extreme weather. The purpose of Figure 4 is to demonstrate that our dataset effectively captures the regions with the greatest prediction loss for each model. These regions correspond to extreme weather events included in our dataset, which was constructed using recorded data and manual filtering. We have also added additional analysis of physical variables in appendix A.1.

---

> > ### Comment · Reviewer_kLu6 · 2024-11-25
> > **Response**
> >
> > Thank you for taking the time to respond to my concerns. The clarifications on the task, and the setup of the NWP model have alleviated many of my concerns. I also appreciated the clarity on the training of each of the baseline models and hyperparameter tuning. I will update my score accordingly

---

> > > ### Author Response · Authors · 2024-11-25
> > > **Thanks**
> > >
> > > We are very glad that our response has addressed your concerns! And thank you very much for your recognition of the value of our paper!

---

### Official Review · Reviewer_Jbrg · 2024-11-03

**Soundness:** 3
**Presentation:** 2
**Contribution:** 3
**Rating:** 8
**Confidence:** 4

**Summary:**

This work utilizes high-resolution HRRR data to create the HR-Extreme dataset, which encompasses a comprehensive set of 17 extreme weather types. The aim is to provide a more specialized dataset for evaluating the performance of weather forecasting models. To achieve this goal, the authors employed unsupervised clustering and manual filtering methods to develop a complete feature map of extreme events in a machine learning-ready format for the continental United States. The dataset was then used to assess the 1-hour forecasting capabilities of existing medium-range prediction models in comparison to the HR-Heim model proposed in this study.

**Strengths:**

1. **Significance**: It fills the gap in benchmarking for extreme weather assessment in deep learning-based medium-range weather forecasting tasks.

2. The dataset is clearly introduced, and it will be fully open-sourced.

3. The authors present comprehensive experiments and baseline results.

**Weaknesses:**

1. The description of the dataset generation process lacks clarity in some areas: a more detailed introduction of the clustering method is needed, including how records from different sources are handled and the hyperparameters of the algorithm.

2. There are also unclear aspects in the experimental description:
   a) The baselines are models trained on globally coarse-resolution grids; Was there any further fine-tuning on this dataset? What preprocessing steps were taken?
   b) What is the training strategy for HR-Heim? Does it use the same hyperparameter settings on the original dataset and HR-Extreme?

**Questions:**

1. Was the result of the clustering algorithm verified by domain experts? Is there corresponding uncertainty detection and assessment?

2. What is the basis for the "types of events without specific ranges or those not related to obvious variations in feature map predictions"?

3. Can data from different sources be used separately?

---

> ### Author Response · Authors · 2024-11-22
> **Response to Reviewer Jbrg [Part 1]**
>
> Thank you for very much your positive feedback ! We truly appreciate your insightful understanding and recognition of the key contributions of our study in terms of significance, comprehensiveness and open-source code and dataset! We have revised our paper according to your suggestions and provide our responses to address your concerns below.
>
> > **Question about clustering method**
>
> **A**: Thank you for your question. We already include the detailed introduction of the clustering method we used (DBSCAN) is in the end of the part "NOAA Storm Prediction Center" in section 3.2,
> but to make it clearer, we improve our writings for it and also put it here for your convenience.
>
> "After extensive case studies, we determined that DBSCAN (Ester et al., 1996) is most suitable for this task, as illustrated in Figure 1. For each timestamp, user reports are treated as 2D points based on normalized latitude and longitude on the x and y axes. DBSCAN identifies clusters based on point density, forming a cluster if there are enough points in close proximity. We carefully tune the hyperparameters of DBSCAN to create more intuitive clusters and to filter out noisy points more accurately as shown in Figure 1. Noisy points are filtered out because they likely represent minor events or errors that are not significant enough to warrant creating a separate cropped area for evaluation."
>
> > **Question about model training**
>
> **A**: Thank you for your question.  All models (Pangu, Fuxi, and our HR-Heim) were trained on HRRR data spanning the U.S. from January 2019 to June 2020, from scratch. They were trained under identical parameters and same level of model parameters, and no hyperparameter tuning was applied to HR-Heim. Furthermore, none of these models were fine-tuned on our HR-Extreme dataset, ensuring a fair basis for comparison and evaluation. This clarification has been added at the beginning of Section 4.3 in blue.
>
> > **Concern about clustering results and more analysis**
>
> **A**: Thank you for your question. We have verified our clustering results for finding the area extreme event by intensive case studies when we built our dataset. We carefully tuned the hyperparameters of it to make the results consistent with the recorded information. We have also added an additional statistical analysis of physical variables in different extreme weather events in appendix A1 showing the distinct characteristics of extreme events compared to normal cases.
>
> > **Question about the sentence: "types of events without specific ranges or those not related to obvious variations in feature map predictions"**
>
> **A**: Thank you for your question. This sentence describes our data cleaning process for entries from the NOAA Storm Events Database. To make it clearer, we have reorganize the sentences as the following and fixed it in Section 3.2 in orange . While most events include information on location and spatial range, some lack these critical details. Additionally, certain event types, such as avalanches and high surf, do not significantly impact the physical variables predicted by the models. To ensure greater accuracy, we have filtered out these events.
>
> >**Can data from different sources be used separately?**
>
> **A**: Yes! The index file generated along with the dataset allows users to easily retrieve information on event types, time spans, and locations for each event from any data source. However, only use a single data source would be incomplete, as the three data sources that we utilize each one contributes unique information on different types of extreme events with varying extents. (more details in Section 3.2)

---

### Official Review · Reviewer_wJpu · 2024-11-09

**Soundness:** 2
**Presentation:** 3
**Contribution:** 2
**Rating:** 5
**Confidence:** 5

**Summary:**

Extreme weather forecasting is a crucial problem for the whole world. With the rise of deep learning-based weather forecasting models, the effectiveness of them on extreme weathers are not well analyzed.  This paper targets on providing a new benchmark for extreme weather forecasting. Authors employ the HRRR data and utilize the extreme events record in three sources to crop the extreme feature from the original HRRR dataset. Experiments are conducted with four baselines to show the performance.

**Strengths:**

Pros:

1. Extreme weather forecasting evaluation is an important research problem.
2. The provided dataset introduces 17 extreme events, which are comprehensive.
3. Authors also release the code for generating the data.

**Weaknesses:**

Cons:

1. It is not clear how the HR-heim model is trained.
2. Considering the dataset is a processed version of HRRR, it would be helpful to provide the geo-location of the extreme data to facilitate more diverse use from users.
3. While the dataset is valuable, there is almost no analysis are present, especially compared to the ERA5 dataset, which is not insightful.

**Questions:**

please refer to the weakness

---

> ### Author Response · Authors · 2024-11-22
> **Response to Reviewer wJpu [Part 1]**
>
> Thank you for your positive comments, and thank you for your recognition of the paper’s importance, comprehensiveness and open-source code and dataset! We have revised our paper according to your suggestions and provide our response to address your concerns below.
> > **Question about how HR-Heim is Trained**
>
> **A**: Thank you for your question. All models (Pangu, Fuxi, and our HR-Heim) were trained on HRRR data spanning the U.S. from January 2019 to June 2020, from scratch. They were trained under identical parameters and same level of model parameters, and no hyperparameter tuning was applied to HR-Heim. Furthermore, none of these models were fine-tuned on our HR-Extreme dataset, ensuring a fair basis for comparison and evaluation. This clarification has been added at the beginning of Section 4.3 in blue.
>
> > **Questions about more metadata in dataset**
>
> **A**: Thank you for your question. This information is already included in the index file. Alongside the dataset generated by our code. The index file contains details on location, range, type of extreme weather, and time span by integrating information from three data sources. Users can easily convert this information from the index file to longitude and latitude coordinates with our open-source code. We have added this clarification at the beginning of Section 3.3 in blue.
>
> > **Questions about more analysis and comparison to ERA5**
>
> **A**: Thank you for your suggestion. We have also added an additional statistical analysis of physical variables in different extreme weather events in appendix A1 showing the distinct characteristics of extreme events compared to normal cases. In terms of ERA5, we have considered ERA5 in our research, however, it is not suitable for building our extreme dataset. ERA5 data has a resolution of 31 km, while HRRR data provides a finer resolution of 3 km. Due to this difference, many extreme events with limited spatial range are more clearly represented in HRRR data but appear vague or may be entirely absent in ERA5. It is more accurate and beneficial for future work to develop our dataset using HRRR data.

---

> > ### Author Response · Authors · 2024-11-29
> > **Response**
> >
> > Thank you once again for your valuable comments and suggestions, which have greatly improved our paper. We have carefully addressed your concerns by revising papers and making clarifications. Could you kindly let us know if you have any further concerns or suggestions?

---

### Author Response · Authors · 2024-11-22
**General Comments and Revision Summary**

We thank all reviewers for your insightful reviews and helpful suggestions, and thank all reviewers for their recognition of the importance, comprehensiveness and contributions of our work.  The revised version of our paper is submitted and we give a summary of our revision of the paper below,:
- According to the reviewer's suggestion, we have added additional statistical analysis of physical variables in normal and extreme events showing the ability of our dataset. We have also added additional experiments of NWP and HR-Heim at more lead times, the results show HR-Heim consistently outperform NWP model at different lead times.
- Many reviewers have concerns about the model training, we have clarified that all deep learning models (Pangu, Fuxi and HR-Heim) are trained with same level of parameters, same hyperparameters. And none of them are fine-tuned on HR-Extreme, enabling fair comparison. This clarification is also added in the paper.
- Some reviewers have questions about more details of extreme events of our dataset, we have clarified that all kinds of details of extreme events can be found in the index file generated along with the dataset. This is clarification is also added to the paper.
- We have improved our writing to present more clearly our primary goals, contributions and significance.
- At last, I want to thank all reviewers again for your constructive feedback that makes this paper better.

---

### Meta-Review · Area_Chair_vFA2 · 2024-12-20

**Metareview:**

The authors introduce a high-resolution weather dataset for extreme weather based on the HRRR public data from NOAA (numerical weather prediction). They do this by processing HRRR through unsupervised clustering and filtering and creating a comprehensive benchmark of many extreme weather events over the US. The data is in ML training ready format and open-sourced. They further introduce a new DL baseline model that outperforms other SOTA DL models on their dataset.

Strengths: Timely dataset for extreme weather forecasting was acknowledged as a strong contribution by all reviewers, dataset is comprehensive, results show the shortcomings for current DL models

Weaknesses: The HR-Heim model contribution as a baseline model is not comprehensive, technical clarity can be improved

The paper could be improved by primarily more discussion/text based on the reviewer's comments and my own reading. For example:

* analysis on ERA5 - The author's rebuttal is satisfactory in that ERA5 is a coarse (but global) dataset. It could be beneficial to show this shortcoming of ERA5 to motivate the new dataset - missing extreme events, smooth/coarse prediction of events, etc through similar visual images.
* NWP comparisons - discussion on why the HR-Heim model is better than NWP given that NWP is used to create the dataset in the first place - this could be a discussion on details of the HRRR dataset (analysis vs reanalysis states), how the NWP model works in the forecasting mode, what additional information the HR-Heim model sees (as well as other DL models) and why you expect this setting is close to the actual forecasting setting for NWP prediction centers.
* HR-Heim vs other models - discussion on why HR-Heim is a better architecture than the other models, given that they are all trained on the same dataset - what parts of HR-Heim contribute to high resolution prediction and what parts could be switched with other backbones. Typically, these are answered systematically with ablations but since the authors reduce this contribution in favor of the dataset primarily, a discussion will be beneficial.
* Model training discussion - while the author's rebuttal clarifies partially how the models were trained, it is beneficial to also state training times, training resources. This also related to the previous point where the compute/memory costs comparisons across models will inform the reader as to the actual cost of getting the increased accuracy on extreme events.
* Minor: Fig resolution for Fig 1 can be improved and in general font sizes for figures can be bigger to allow for easy readability.

The above improvements (primarily text changes) will help cater to the broader ML for science ICLR audience.

**Additional Comments On Reviewer Discussion:**

There were multiple concerns on details about how the models were trained which the authors have answered mostly.
The other questions (see improvement list above) were also raised and answered in the rebuttal but I believe they should be included as a discussion in the text of the paper as well.

wJpu had the least opinion and raised 3 weaknesses on model training, geo-location info in the data, and comparisons to ERA5. I believe the authors have answered the first two (see above for partial clarification on model training); for ERA5 comparisons, there could be more discussion - see improvement list above.

The reviewers were very positive on the dataset contribution. I weight some of the technical discussion gap more heavily for a complete understanding of the dataset as well as models used. I believe they can be addressed with textual changes and more information (no new experiments) which the authors mostly have in their rebuttal and hence still lean towards acceptance.

---

### Decision · Program_Chairs · 2025-01-22

Accept (Poster)